# Tolerance of Honey Bees to *Varroa* Mite in the Absence of Deformed Wing Virus

**DOI:** 10.3390/v12050575

**Published:** 2020-05-23

**Authors:** John M. K. Roberts, Nelson Simbiken, Chris Dale, Joel Armstrong, Denis L. Anderson

**Affiliations:** 1Commonwealth Scientific and Industrial Research Organisation, Canberra 2601, Australia; joel.armstrong@csiro.au; 2Coffee Industry Corporation Ltd., Goroka 441, Papua New Guinea; nsimbiken@cic.org.pg; 3Department of Agriculture, Water and the Environment, Canberra 2601, Australia; Chris.Dale@awe.gov.au; 4Research and Development Division, Abu Dhabi Agriculture & Food Safety Authority, Al Ain, UAE; denis.l.anderson@ADAFSA.GOV.AE

**Keywords:** *Apis*, *Varroa jacobsoni*, pollinator, virus discovery, iflavirus, RNA viruses, next-generation sequencing

## Abstract

The global spread of the parasitic mite *Varroa destructor* has emphasized the significance of viruses as pathogens of honey bee (*Apis mellifera*) populations. In particular, the association of deformed wing virus (DWV) with *V. destructor* and its devastating effect on honey bee colonies has led to that virus now becoming one of the most well-studied insect viruses. However, there has been no opportunity to examine the effects of Varroa mites without the influence of DWV. In Papua New Guinea (PNG), the sister species, *V. jacobsoni*, has emerged through a host-shift to reproduce on the local *A. mellifera* population. After initial colony losses, beekeepers have maintained colonies without chemicals for more than a decade, suggesting that this bee population has an unknown mite tolerance mechanism. Using high throughput sequencing (HTS) and target PCR detection, we investigated whether the viral landscape of the PNG honey bee population is the underlying factor responsible for mite tolerance. We found *A. mellifera* and *A. cerana* from PNG and nearby Solomon Islands were predominantly infected by sacbrood virus (SBV), black queen cell virus (BQCV) and Lake Sinai viruses (LSV), with no evidence for any DWV strains. *V. jacobsoni* was infected by several viral homologs to recently discovered *V. destructor* viruses, but Varroa jacobsoni rhabdovirus-1 (ARV-1 homolog) was the only virus detected in both mites and honey bees. We conclude from these findings that *A. mellifera* in PNG may tolerate *V. jacobsoni* because the damage from parasitism is significantly reduced without DWV. This study also provides further evidence that DWV does not exist as a covert infection in all honey bee populations, and remaining free of this serious viral pathogen can have important implications for bee health outcomes in the face of Varroa.

## 1. Introduction

Pathogens have a major impact on honey bee (*Apis mellifera*) populations, with none more so than the parasitic mite *Varroa destructor* and several associated viruses [1,2,3]. Since shifting from its original Asian honey bee (*A. cerana*) host last century, *V. destructor* has spread globally, causing significant harm to *A. mellifera* populations by feeding on developing pupae and adult bees. Importantly, this host shift also led to the emergence of virulent iflavirus strains of deformed wing virus (DWV), of which there are three known master variants (DWV-A, DWV-B and DWV-C) that are effectively vectored by *V. destructor* [4]. Several studies have reported on this emergence following the spread of *V. destructor,* and from this it has been generalised that DWV exists latently in all *A. mellifera* populations [5,6]. However, evidence supporting the absence of DWV in the Australian *A. mellifera* population, which has yet to be invaded by parasitic bee mites, and the absence of DWV from New Zealand until the arrival of *V. destructor* [7], contradicts this theory and suggests that DWV is not ubiquitous in all honey bee populations [8,9]. The near worldwide distributions of *V. destructor* and DWV have left few opportunities to examine their individual impacts [10]. However, Papua New Guinea (PNG), a large Pacific island to the north of Australia, is home to a unique *A. mellifera* population that does not have *V. destructor* but has recently experienced a host shift from the sister mite species *V. jacobsoni* [11]. This new mite threat was detected in 2008, causing colony losses for local beekeepers and pathologies consistent with parasitic mite syndrome [12]. However, it is not clear if the effects of this new mite are also associated with viruses.

Honey bees (*Apis* spp.) are not native to PNG. The two species currently found there, *A. mellifera* and *A. cerana*, were introduced in the last 80 years through human activities. European honey bees (*A. mellifera*) were first introduced in the late 1940s from Australia [13,14], and later through introductions from New Zealand and Australia as part of bilateral aid projects [15,16,17]. Hence, all present-day *A. mellifera* in PNG are descendants of bees originally imported from Australia and New Zealand. *Apis cerana* were first introduced into Indonesian Papua (western half of New Guinea) from Indonesian Java during the 1970s with human transmigration [18]. Wild colonies subsequently spread and established throughout the island and into neighbouring PNG. The introduction of *A. cerana* also brought the parasitic mite, *V*. *jacobsoni,* of which it is the native host [19,20]. This mite was restricted to *A. cerana* for ca. three decades, invasively spreading with *A. cerana* to nearby Solomon Islands in 2003 [21] and finally adapting a parasitic reproducing form on *A. mellifera* in PNG by 2008. After initial colony losses, PNG beekeepers were reportedly managing to withstand *V. jacobsoni* infestations without any chemical or cultural control (pers. comm J. Buka IHBA).

Soon after the emergence of pathogenic *V. jacobsoni*, PNG was also invaded by a second parasitic mite, *Troplilaelaps mercedesae,* originating from Indonesia [12,18]. DWV has been reported to be associated with this mite in other regions of the world [22,23]. The combination of the two mites once again increased colony losses, and in 2016 (8 years after the reproducing form of *V. jacobsoni* was discovered) the miticide Bayvarol was used for the first time in PNG in efforts to eradicate *T. mercedesae* [24]. This was ultimately unsuccessful and the use of Bayvarol has essentially stopped due to the high cost for local beekeepers. Despite the impact of these two significant mite parasites and apparently similar infestation levels to that of *V. destructor*, PNG beekeepers have continued to persist without any targeted mite management [24]. This suggests that there is some unknown mechanism that allows the PNG *A. mellifera* population to better resist the impact of parasitic mites.

In this study, we investigated if the mechanism conferring tolerance to *V. jacobsoni* is related to a different viral landscape compared with *V. destructor*-affected honey bee populations. There is very little knowledge of the viruses infecting honey bees in PNG, although previous serological testing of *A. mellifera* colonies identified the common bee viruses, sacbrood virus (SBV), black queen cell virus (BQCV), Kashmir bee virus (KBV) and chronic bee paralysis virus (CBPV) [16]. We hypothesised that because the *A. mellifera* in PNG were originally imported from Australia, which is free of DWV, and New Zealand before DWV was present there, perhaps this population is also not infected by strains of DWV, and hence their absence may confer a level of tolerance toward *V. jacobsoni*. To assess this, we used a combination of high-throughput sequencing (HTS) and reverse-transcription PCR to identify RNA viruses infecting the bees and mites in PNG and bees in nearby Solomon Islands.

## 2. Materials and Methods

### 2.1. Sample Collection

Female *V. jacobsoni* that were reproducing on *A. mellifera* were collected during their initial discovery in 2008 from brood cells of two different *A. mellifera* colonies in Goroka, PNG. Mites were collected in RNAlater (Ambion, Inc., Austin, TX, USA) and stored at −20 °C.

Samples of *A. mellifera* and *A. cerana* worker bees were collected from colonies across the PNG Highlands in 2014, 2015 and 2018, and from Solomon Islands in 2014 (Appendix A). A combined sample of at least 50 bees was collected from multiple hives (if present) at each apiary location. These samples were collected in 70–100% ethanol for transport to the laboratory and stored at −20 °C.

### 2.2. Sample Processing for Viral Analysis

Total RNA was extracted from 20 pooled adult female mites for each colony using a standard Trizol reagent (ThermoFisher Scientific, Waltham, MA, USA) extraction protocol, except for precipitated RNA, which was left in 70% ethanol. The samples were then purified using the Purelink RNA mini kit (ThermoFisher Scientific, Waltham, MA, USA) and on-column Purelink DNase treatment protocol (ThermoFisher Scientific, Waltham, MA, USA) before being eluted in 50 µL of RNase-free water. Illumina mRNA libraries were prepared from the extracted RNA and 100 bp paired-end sequences were generated on an Illumina HiSeq 2000 (Illumina, San Diego, CA, USA).

*A. mellifera* samples collected in May 2014 from PNG and September 2014 from Solomon Islands were used for HTS with viral enrichment steps. First, 50 worker bees per apiary location were macerated in 5 mL of 0.05 M potassium phosphate buffer in an extraction bag (Bioreba AG, Kanton Reinach, Switzerland). Two pooled samples were created using 1 mL of each apiary sample (PNG *n* = 14, Solomon Islands *n* = 9) into a screw-cap centrifuge tube and adjusted with 0.05 M potassium phosphate buffer for a total volume of 20 mL. 3 mL diethyl ether and 3 mL chloroform were added, shaken vigorously and centrifuged at 6000 rpm for 30 min (J-E Avanti centrifuge, Beckman Coulter, Brea, CA, USA). Supernatants were transferred to Ultraclear SW28 tubes (Beckman Coulter, Brea, CA, USA) and centrifuged at 21,500 rpm for 3.5 h at 4 °C (Beckman L-80 ultracentrifuge, Beckman Coulter, Brea, CA, USA). Pelleted samples were dissolved in 1 mL of 0.05 M potassium phosphate buffer, then passed through a 0.22 µm filter to remove bacterial contamination. A total of 340 µL of each filtered sample was mixed with 10 µL of RNase, 10 µL DNase and 40 µL of DNase I buffer and incubated at 37 °C for 2 h. RNA was extracted from the treated samples using the Purelink viral RNA extraction kit (ThermoFisher Scientific, Waltham, MA, USA). Illumina mRNA libraries were prepared and 100 bp paired-end sequences were generated on an Illumina HiSeq 2500 (Illumina, San Diego, CA, USA).

*A. mellifera* (11 samples) and *A. cerana* (six samples) samples collected in November 2015 across the PNG Highlands were tested by RT-PCR for DWV. As above, 30 worker bees per apiary location were macerated in 5 mL of 0.05 M potassium phosphate buffer in an extraction bag (Bioreba AG, Kanton Reinach, Switzerland). RNA was extracted from 200 µL of sample using the Purelink viral RNA/DNA extraction kit (ThermoFisher Scientific, Waltham, MA, USA) and tested by real-time RT-PCR for DWV-A and DWV-B using the primers developed by Kevill et al. [25]. One-step real time RT-PCR reactions were carried out in duplicate for each RNA sample using SensiFAST™ SYBR^®^ No-ROX One-Step mix (Bioline, London, UK) alongside synthetic positive controls for the target region of DWV-A and DWV-B produced by GeneArt™ (ThermoFisher Scientific, Waltham, MA, USA). RNA quality of each sample was also confirmed by amplification of the honey bee gene vitellogenin [26].

*A. mellifera* (13 samples) and *A. cerana* (three samples) samples collected in May and October 2018 from PNG were tested by RT-PCR for DWV and also used for HTS with only limited viral enrichment. As above, 30 worker bees per apiary location were macerated in 5 mL of 0.05 M potassium phosphate buffer in an extraction bag (Bioreba AG, Kanton Reinach, Switzerland) and 200 µL used for RNA extraction for real-time RT-PCR. In addition, 1 mL of each sample was centrifuged at 17,000 *g* for 3 min and the supernatant passed through a 0.45 µm filter as suggested by Conceição-Neto et al. [27] for viral enrichment/bacterial reduction. Samples were mixed with an equal volume of Purelink viral lysis buffer (ThermoFisher Scientific, Waltham, MA, USA) and 20 µL proteinase K and incubated at 56 °C for 1 h. Lysed samples were each passed through the extraction spin column of the Purelink viral RNA/DNA extraction kit (ThermoFisher Scientific, Waltham, MA, USA) to create two pooled samples for *A. mellifera* and *A. cerana*. Total RNA libraries were made and 50 bp paired end sequences were generated on the NovaSeq 6000 (Illumina, San Diego, CA, USA). Library preparation and sequencing was carried out at the Biomedical Research Facility (Australian National University, Canberra).

### 2.3. Viral sequence analysis

Sequence analysis was carried out with CLC Genomics Workbench v11 (CLC Bio, Aarhus, Denmark) with raw data first quality trimmed and adapter sequences removed before trimmed reads were mapped to reference host genomes from the NCBI GenBank database. Unmapped reads ranged from 27% to 37% for *V. jacobsoni* libraries and between 53% and 71% for honey bee libraries receiving viral particle enrichment. Unmapped reads were collected and then mapped to the virus reference genome database retrieved from NCBI using a length fraction of 0.5 and similarity fraction of 0.8. Consensus sequences were manually inspected for genome coverage and similarity to mapped reference genomes using BLASTn. Unmapped reads were also de novo assembled in CLC Genomics workbench using default parameters and a minimum contig length of 1000 nucleotides. Contigs were compared to the NCBI virus reference genome database using BLASTn to identify known viruses and compared to the NCBI non-redundant protein database using BLASTx to identify putative novel virus genomes. Sequence alignments, annotation and phylogenetic analysis of viral genomes were done in Geneious v2020.0.5 (Biomatters Ltd., Auckland, New Zealand). Raw data and virus sequences have been submitted to GenBank (MT482464-MT482499).

## 3. Results

### 3.1. HTS Virus Detection in V. jacobsoni Reproducing on A. mellifera

Our first opportunity to explore the viruses associated with the new pathogenic *V. jacobsoni* in PNG, was to undertake HTS of two pooled samples of pathogenic *V. jacobsoni* collected in 2008. Sequence reads were first mapped to Varroa reference genomes and the unmapped reads collected for viral analysis by mapping to the NCBI virus reference database and de novo assembly with contigs BLAST searched for similarity to known viruses. These two approaches revealed that homologous isolates to several known Varroa-associated viruses were present in these samples (Table 1). Two rhabdoviruses were found which appeared to be homologs of ARV-1 and ARV-2 in only sharing 73% and 64% amino acid identity with the *V. destructor* isolates. We have designated these novel variants as Varroa jacobsoni rhabdovirus 1 and 2 (VJRV-1 and VJRV-2). Several contigs in both samples were most similar to the novel iflavirus VDV-2 with partial genomes for two major variants emerging. Consensus sequences for these variants shared ca 96% amino acid identity across the polyprotein and were between 88% and 92% similar to the VDV-2 ISR and UK isolates. These viruses were dominant in both mite samples and are named here as Varroa jacobsoni virus-2 (VJV-2), PNG isolate 1 and 2. Both samples also had genome length contigs with 84% amino acid identity to VDV-4, a novel unclassified +ssRNA virus detected in *V. destructor* on *A. cerana* from Thailand [28]. Lastly, four short contigs ranging from 1700 to 3700 nucleotides were detected with homology to ssDNA circoviruses. Importantly, there was no evidence for any known honey bee viruses in either mite sample.

### 3.2. HTS Virus Detection in A. mellifera in PNG and Solomon Islands 2014

Our next opportunity was to examine the viral landscape of honey bees in the Pacific using HTS of two pooled *A. mellifera* samples collected across PNG in May 2014 and Solomon Islands in September 2014. Read mapping to the honey bee genome and de novo assembly of the unmapped reads revealed the known honey bee viruses, SBV, BQCV, IAPV and LSVs in both samples (Table 1). SBV was by far the most abundant virus in both samples, although each possessed a distinct variant. The SBV variant present in PNG was the same as previously reported by the authors and referred to as the PNG serotype [29]. This variant is ca. 10% different to other SBV variants circulating in *A. mellifera* and *A. cerana*. However, the SBV variant detected in Solomon Islands matched to Australian isolates of the *A. mellifera* strain. The four LSV strains detected were also most similar to Australian isolates.

Notably there was no genetic evidence for any DWV variants in either sample. However, single contigs representing partial genome sequences were present in each sample with distant similarity to DWV (Figure 1). The PNG contig was 2063 nucleotides in length, covering the conserved domain of the RNA-dependent RNA polymerase and shared only ca. 59% identity to DWV master variants. The Solomon Islands contig was 4799 nucleotides in length spanning the viral capsid proteins and RNA helicase domains. This contig was ca. 66% similar to DWV strains, although ca. 85% identity was found over the highly conserved RNA helicase domain. Interestingly, this contig was ca. 94% similar to the novel iflavirus Bundaberg bee virus 6 that was recently identified from Australian *A. mellifera* [8].

Several putative novel virus genomes were found in both PNG and Solomon Islands *A. mellifera* (Table 2). A full genome sequence was recovered for a putative novel virus from PNG *A. mellifera* which was the most abundant virus following SBV and BQCV. This virus, named PNG bee virus 1, had a calici-like coat protein and genome organization, suggesting it is an unclassified norovirus of the Caliciviridae. Twelve partial virus genomes were also detected from PNG *A. mellifera* including several putative Dicistroviridae, Iflaviridae and one other Caliciviridae virus. A further two distinct partial virus genomes were recovered from Solomon Islands *A. mellifera*. One was a putative novel iflavirus and the other was 74% similar to Riptortus pedestris virus-1, a novel virus of bean bug that is a pest in Solomon Islands.

### 3.3. HTS Virus Detection in A. mellifera and A. cerana in PNG 2018

Lastly, we used HTS to analyse two pooled honey bee samples of *A. mellifera* and *A. cerana* collected from PNG in May and October in 2018. Once again, reference mapping and de novo assembly of the unmapped reads revealed the known honey bee viruses, SBV, BQCV, IAPV and LSVs in both samples but no DWV variants (Table 1). The *A. cerana* sample was dominated by SBV (*A. cerana* strain) and only low detection of other bee viruses. The *A. mellifera* sample had a similar viral profile to the 2014 sample but also had low presence of VJV-1, which would have been excluded from the 2014 samples due to the viral enrichment method used affecting enveloped viruses. Few novel virus sequences were detected from these samples compared with the 2014 samples, which we suspect is also due to the different enrichment approaches and shorter (50 bp) library sequencing. Two partial RNA virus genomes were found in *A. mellifera* and one partial virus genome was found in *A. cerana.* Interestingly, one of the virus sequences in *A. mellifera* was identical to the PNG bee virus 9 detected in the 2014 sample, suggesting this virus is a persistent infection. The other virus sequence in *A. mellifera* was identical to the virus sequence recovered from *A. cerana*. This putative novel virus (PNG bee virus 14) appears to be an Alphanodavirus of the Nodaviridae, which have a bipartite genome. A Noda-like virus was also reported from *A. mellifera* in New Zealand, although no sequence information is available for comparison [30].

### 3.4. Diagnostic RT-PCR for DWV in A. mellifera and A. cerana

In addition to HTS, we also screened for DWV-A and DWV-B by real-time RT-PCR in a total of 48 *A. mellifera* and 14 *A. cerana* samples collected between 2014 and 2018 from the PNG Highlands and Solomon Islands. In all cases, there was no evidence of either DWV strains infecting any *A. mellifera* or *A. cerana* sample.

## 4. Discussion

Our study brings together viral data from over a decade since *V. jacobsoni* shifted hosts to become pathogenic to *A. mellifera* in PNG. Using HTS and real-time RT-PCR viral analysis, we demonstrated that the viral profiles of *V. jacobsoni, A. mellifera* and *A. cerana* in PNG and *A. mellifera* in Solomon Islands have several common honey bee viruses but not any strain of DWV. This is very different from most other honey bee populations around the world, were DWV has become ubiquitous alongside *V. destructor* [31]. The unique PNG case offers a rare view of the impact of parasitic mites without the confounding effects of DWV. Other studies have investigated the effects of DWV in vitro [32,33,34,35], but the widespread nature of the virus has made it difficult to independently observe the direct effects of *Varroa* parasitism on colony health. Our findings suggest that the absence of DWV in PNG is a key underlying factor in the apparent tolerance of this *A. mellifera* population to *V. jacobsoni*. Multiple resistance traits have evolved in different *A. mellifera* populations around the world that suppress mite levels through honey bee behaviours [36,37] or confer tolerance to high DWV titres [38,39]. It is unknown what tolerance traits (if any) have developed in PNG to allow local beekeepers to overcome initial colony losses and maintain colonies without mite management. It is possible that the damage from mites is simply low without the added pathogen pressure of DWV and other important brood diseases like chalkbrood and European foulbrood [9,16]. Indeed, modelling approaches suggest that honey bee colonies can tolerate substantially higher mite (*V. destructor*) populations in the absence of DWV [40]. A similar scenario is suggested for the Varroa-tolerant honey bees of Fernando de Noronha, where DWV levels have remained very low, thus limiting the impact on colonies [41]. However, this population also had very low reproductive rates for *V. destructor*, suggesting that additional tolerance mechanisms may be involved. This may also be the case for *V. jacobsoni* in PNG, with potentially lower virulence to *A. mellifera* than its sister species *V. destructor*. Further investigations are still needed in PNG to better understand the impacts of *V. jacobsoni* and *T. mercedesae* on colony health and identify potential tolerance traits that have evolved.

The absence of DWV in PNG and Solomon Islands is in stark contrast with the currently held belief that this virus exists in all *A. mellifera* populations as a covert infection until *V. destructor* disrupts this balance and allows virulent strains in the bee to proliferate [6]. This is not the first evidence to cast doubt on this idea, with the Australian *A. mellifera* population recently shown to be free of DWV strains [8,9,35,42,43]. Surveys of other honey bee populations in the Pacific where *Varroa* mites have not reached have also not detected DWV (Norfolk Island [44]; Fiji [45]; Vanuatu [46]; Samoa [47]). In fact, only two countries in the Australia-Pacific region are confirmed to have DWV: New Zealand and the Kingdom of Tonga, and both are likely the result of recent introductions. Following the arrival of *V. destructor* in New Zealand, DWV was not initially detected [7] but was later found by Mondet et al. [5] having displaced other honey bee viruses along the *V. destructor* expansion front. However, rather than the emergence of a long-present covert infection, it is likely that the authors observed the establishment of DWV that arrived with or after the arrival of *V. destructor*. In the Kingdom of Tonga, *V. destructor* were introduced in 2006 through the importation of queen bees from New Zealand [48]. This presumably also introduced DWV, which by this time was circulating in New Zealand. Therefore, as *A. mellifera* in these Pacific Island countries are descendants of Australian and New Zealand stock [49], it appears the broader region was originally naïve of DWV. With PNG as a case study for the effects of Varroa without DWV, biosecurity authorities should aim to prevent further introductions of this pathogenic virus to the Australia-Pacific region.

Interestingly, partial genomes for two putative novel viruses were recovered from *A. mellifera* in PNG and Solomon Islands that showed low similarity to DWV strains. The virus sequence found in Solomon Islands was closely related (94%) to the recently identified Bundaberg bee virus 6, whereas the PNG virus sequence represents a third DWV-like virus detected from this region—with Bundaberg bee virus 6 and Darwin bee virus 3 being the other DWV-like viruses detected from Australian *A. mellifera* [8]. The relevance of these DWV-like viruses is intriguing as they could have the same potential to become pathogenic in Varroa-infested honey bee populations. It is clear that slow-replicating viruses, like DWV, will more likely become emerging pathogens with the aid of *V. jacobsoni* [5,35,50]. Therefore, the isolation and characterisation of these DWV-like viruses would be valuable to compare virulence with DWV strains. While there was no evidence that this DWV-like virus or any other putative novel viruses were emerging in association with *V. jacobsoni*, there were several viruses with relatively high abundance compared with the known honey bee viruses in PNG which may also warrant further investigation.

There was almost no overlap in the viral profiles of *V. jacobsoni* and honey bees in PNG or Solomon Islands. Aside from DWV, other honey bee viruses have been commonly detected in *V. destructor* [51,52]. In our study, we found no honey bee viruses in *V. jacobsoni,* despite the high abundance of SBV in PNG *A. mellifera* and *A. cerana*. The only shared virus was VJRV-1, which is homologous to ARV-1, and found in low abundance in the PNG *A. mellifera* 2018 sample. ARV-1 has been shown to replicate in both *A. mellifera* and *V. destructor* and can spill over into bumblebees [48,53]. However, our findings suggest VJV-1 and VJV-2 predominantly infect *V. jacobsoni* and possibly do not replicate in *A. mellifera*. VJV-1 and VJV-2 are quite divergent from ARV-1 and ARV-2, with less than 73% amino acid identity, and could feasibly have a different ecology. Interestingly, ARV-2 was found to be common in Thai *A. cerana*, whereas no Varroa-associated virus was detected in *A. cerana* in PNG. We also detected a *V. jacobsoni* homolog (VJV-4) of the virga-like virus recently discovered infecting *V. destructor* in Thailand [28], suggesting this virus may be common in Varroa mites throughout Asia. VJV-4 and VDV-4 also cluster phylogenetically with several novel viruses sequenced from spiders in China and USA and ticks in Australia [54,55], indicating the evolution of an arachnid clade for this diverse virga-like virus group.

The two iflavirus VJV-2 isolates (VDV-2 homologs) were the most abundant viruses in *V. jacobsoni*, which is consistent with the findings of Herrero et al. [56] and Levin [57], who found VDV-2 to be prevalent in *V. destructor* from multiple geographic populations. Herrero et al. [56] also found VDV-2 in *A. mellifera* pupae, but suggest this was contamination from feeding mites and not active infections. In our study, we did not detect VJV-2 in adult honey bees nor did Levin [57] when they screened colonies for VDV-2, indicating that these are probably Varroa-specific viruses that do not persist in *A. mellifera*. Interestingly, we found two VJV-2 strains co-occurring in *V. jacobsoni* at similarly high levels. In contrast, only single strains of VDV-2 (Israel and UK strains) have been reported infecting *V. destructor*. Potentially, these two strains represent master variants similar to that described for the honey bee iflaviruses DWV, SBV and slow bee paralysis virus (SBPV), and perhaps the presence of two master variants is a common feature of invertebrate iflaviruses. This mirroring of viral profiles between two Varroa species is also interesting and suggests a long association that may extend back to a common ancestor. Determining the prevalence and pathology of these *V. jacobsoni* viruses will help to understand their importance and provide useful comparative insights with *V. destructor.*

This study has highlighted a unique mechanism for Varroa tolerance in PNG *A. mellifera*, where the absence of DWV appears to have limited the impact of *V. jacobsoni* on colony health. However, this also suggests that there is potential for a virulent viral pathogen to emerge in this population, whether from the introduction of DWV or the emergence of a novel virus that can exploit this niche. Therefore, the impact of *V. jacobsoni* could change over time and should be monitored, alongside any changes to the viral landscape. Determining whether other Varroa tolerance mechanisms have also evolved or if there are any biological differences that reduce the virulence of *V. jacobsoni* compared with *V. destructor* also have valuable implications for understanding this novel host–pathogen relationship that has emerged in PNG.

## Figures and Tables

**Figure 1 viruses-12-00575-f001:**
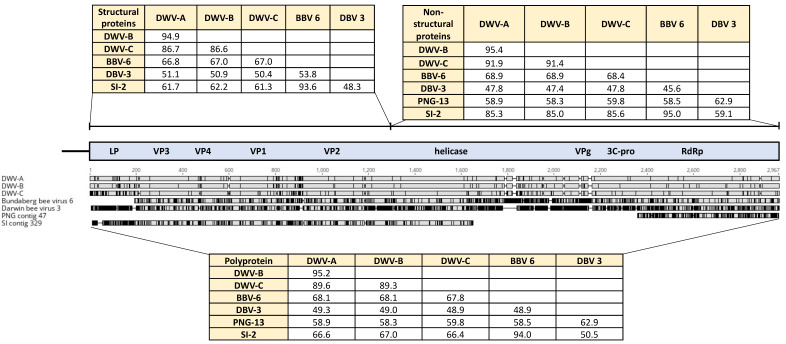
Amino acid alignment of novel virus sequences detected in Australia and the Pacific with DWV master variants. Percentage amino acid identities for the structural (leader protein, LP; viral coat proteins, VP) and non-structural (helicase; viral genome-linked protein, VPg; 3C protease, 3C-pro; RNA-dependent RNA polymerase, RdRp) protein regions of the viral genome are shown. BBV-6, Bundaberg bee virus 6; DBV-3, Darwin bee virus 3; PNG-13, PNG bee virus 13; SI-2 SI bee virus 2.

**Table 1 viruses-12-00575-t001:** Transcripts per million (TPM) for detected honey bee and Varroa viruses.

Viruses Detected	*Vd* Homolog	*Vj-*B1 2008	*Vj-*C1 2008	PNG *Am* 2014	Sol. Is. *Am* 2014	PNG *Am* 2018	*PNG Ac* 2018
VJRV-1 (MT482464)	ARV-1	24,476	29,575	0	0	31	0
VJRV-2 (MT482465)	ARV-2	34,735	77,497	0	0	0	0
VJV-2 PNG1 (MT482466)	VDV-2	199,968	130,682	0	0	0	0
VJV-2 PNG2 (MT482467)	VDV-2	511,272	407,942	0	0	0	0
VJV-4 (MT482468)	VDV-4	229,548	354,301	0	0	0	0
SBV	na	0	0	705,914	886,074	36,903	379,433
BQCV	na	0	0	32,943	8680	7028	55
IAPV	na	0	0	15	21	5	0
LSV-1	na	0	0	2620	0	66,617	0
LSV-2	na	0	0	8628	43,407	45,055	0
LSV-3	na	0	0	7473	0	91,150	82
LSV-8	na	0	0	176,038	0	272,652	215

*Vd, V. destructor; Vj, V. jacobsoni; Am, A. mellifera; Ac, A. cerana;* VJRV, Varroa jacobsoni rhabdovirus; VJV, Varroa jacobsoni virus; SBV, sacbrood virus; BQCV, black queen cell virus; LSV, Lake Sinai virus.

**Table 2 viruses-12-00575-t002:** Novel virus genomes detected in honey bees from PNG and Solomon Islands.

Novel Virus	Length	Taxonomy	Closest Relative	% Amino Acid Identity	GenBank Accession
**PNG 2014**					
PNG bee virus 1	10,129	Caliciviridae	Hubei picorna-like virus 68	29	MT482483
PNG bee virus 2	7385	Dicistroviridae	Melipona quadrifasciata virus 1a	38	MT482484
PNG bee virus 3	4258	Dicistroviridae	Plautia stali intestine virus	61	MT482485
PNG bee virus 4	4236	unclassified	Hubei picorna-like virus 51	43	MT482486
PNG bee virus 5	4084	unclassified	Bundaberg bee virus 8	28	MT482487
PNG bee virus 6	4077	unclassified	Hubei arthropod virus 1	37	MT482488
PNG bee virus 7	4017	unclassified	Darwin bee virus 6	44	MT482489
PNG bee virus 8	3854	Dicistroviridae	Solenopsis invicta virus 13	51	MT482490
PNG bee virus 9	3738	Caliciviridae	Thika virus	45	MT482491
PNG bee virus 10	3483	Iflaviridae	La Jolla virus	45	MT482492
PNG bee virus 11	3082	Dicistroviridae	Darwin bee virus 6	70	MT482493
PNG bee virus 12	3065	Caliciviridae	Hubei picorna-like virus 67	24	MT482494
PNG bee virus 13 *	2063	Iflaviridae	Darwin bee virus 3	62	MT482495
**Solomon Islands 2014**					
SI bee virus 1	5882	Iflaviridae	Bradson virus	55	MT482497
SI bee virus 2 *	4799	Iflaviridae	Bundaberg bee virus 6	94	MT482498
SI bee virus 3	4280	unclassified	Riptortus pedestris virus-1	74	MT482499
**PNG 2018 samples**					
PNG bee virus 9	4567	unclassified	Thika virus	45	MT482491
PNG bee virus 14 (*A. mellifera)*	3137	Nodaviridae	Mosinovirus–RNA1	42	MT482496
PNG bee virus 14 (*A. cerana*)	2115	Nodaviridae	Mosinovirus–RNA1	42	MT482496

* DWV-like viruses.

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
