# Peer review of "Tolerance of Honey Bees to Varroa Mite in the Absence of Deformed Wing Virus"

_viruses, 2020, doi:10.3390/v12050575_

Round 1
Reviewer 1 Report
This important study investigates a very rare situation of populations of honeybees that are infested with varroa mites but are free of DWV. This is only the second study to look at this rare situation (see Brettell & Martin, 2017), but this is the first to look at the situation in V. jacobsoni, the sister species of V. destructor that has spread around the world killing so many colonies. It is very important study since it provides a vital insight to the role of DWV in killing millions of colonies around the globe. Furthermore, these honeybee populations are in remote locations where the senior author has a long history of study so their history is well understood. Furthermore, despite the presence of the varroa mite these populations have evolved tolerance towards it. Naturally evolved varroa tolerance/resistance is currently a major topic in the fields of honeybee virology and biology, so it will become well cited. The paper is well written, using well established methods and the data supports their conclusions.
There are two (one recent) papers that are highly relevant to this study that need discussed in light of the similarity of there findings and suggest a direction for possible research into discovering their tolerance mechanism.
I only have one minor but important point; near the start of the second paragraph of the discussion there is a statement ‘V. destructor disrupts this balance and causes virulent strains to proliferate [6]’ . This sentence is slightly misleading, since the appearance of a virulent strain appears in the bee, not the mite (Ryabov et al 2014). The presence of the mite is in no way involved with the activation of DWV (see Brettell & Martin 2017), but needs to encounter a bee/colony which has suffered a natural covert DWV infection, a rare event, only then can a new bee-bee transmission route of DWV start.
Brettell LE, and Martin SJ (2017) Oldest Varroa tolerant honey bee population provides insight into the origins of the global decline of honey bees. Scientific Reports 7:45953 DOI: 10.1038/srep45953.
[A studied a varroa infested tolerant population on a remote island that also had no DWV]
Martin SJ, Hawkins GP, Brettell LE, Reece N, Correia-Oliveira ME, Allsopp MH (2019) Varroa destructor reproduction and cell re-capping in mite-resistant Apis mellifera populations. Apidologie DOI: 10.1007/s13592-019-00721-9
[This studies natural tolerant honeybee populations in several countries and finds the same behavioural traits]
Author Response
We thank reviewer 1 for their encouraging feedback on our manuscript and have addressed their comments.
1. There are two (one recent) papers that are highly relevant to this study that need to be discussed; Brettell LE, and Martin SJ (2017) and Martin et al 2019.
Thank you for this suggestion. We had overlooked the relevance of these papers to our study and have now included them in our discussion in the first paragraph.The Brettell & Martin 2017 paper is particularly relevant, having implicated very low DWV as a likely explanation for the Varroa tolerance observed on Fernando de Noronha.
2. ‘V. destructor disrupts this balance and causes virulent strains to proliferate [6]’ . This sentence is slightly misleading, since the appearance of a virulent strain appears in the bee, not the mite (Ryabov et al 2014). The presence of the mite is in no way involved with the activation of DWV (see Brettell & Martin 2017), but needs to encounter a bee/colony which has suffered a natural covert DWV infection, a rare event, only then can a new bee-bee transmission route of DWV start.
We have amended this statement to be more precise and now states "The absence of DWV in PNG and Solomon Islands is in stark contrast with the currently held belief that this virus exists in all A. mellifera populations as a covert infection until V. destructor disrupts this balance and allows virulent strains in the bee to proliferate [6].
However, we do not necessarily agree that Varroa has no involvement with DWV activation, or that it requires a natural overt DWV infection to occur before the mite transmission route can occur. The Brettell & Martin (2017) study only present a possible explanation for the tolerance of bees on Fernando de Noronha. While having merit, this has not been demonstrated.
Reviewer 2 Report
It is very interesting to learn about a host parasite relationship between honeybees and parasitic Varroa mites, without the interference with the DWV complex. I am not a virologist, but very much interested in the evolution of the H-P relationship including DWV. Much can be learned having a case without DWV, while in the presence both resistance mechanisms (lowered population growth) as well as tolerance mechanisms (lower damage despite high population, possibly through resistance or tolerance for viruses) appear to be involved.
I have very few comments: Table 2: what is the meaning of this asterisk to this figure?
PNG bee virus 1 |
10,129* |
Caliciviridae |
......important brood diseases like chalkbrood and European foulbrood [9,16]. It may also be that V. jacobsoni is less virulent to A. mellifera than its sister species V. destructor. Further investigations are still needed in PNG to better understand the impacts of V. jacobsoni and T. mercedesae on colony health and identify potential tolerance traits that have evolved.
- Is something known about population sizes of V jacobsoni in A mellifera colonies? Are these similar / as high as those of V destructor?
Author Response
We thank reviewer 2 for reading our manuscript and for providing the positive feedback. We have addressed the comments below.
1. Table 2: what is the meaning of this asterisk to this figure?
Thank you for picking this up. We were initially going to mark this viral sequence as it was a full genome sequence - all others were partial genomes sequences - but decided this was not needed. Asterisk is now removed.
2. Is something known about population sizes of V jacobsoni in A mellifera colonies? Are these similar / as high as those of V destructor?
Anecdotally V. jacobsoni levels appear similar to V. destructor, but there is little empirical data. We do have some data from another recent study generally supporting this [Roberts, J. M., Schouten, C. N., Sengere, R. W., Jave, J., & Lloyd, D. (2020). Effectiveness of control strategies for Varroa jacobsoni and Tropilaelaps mercedesae in Papua New Guinea. Experimental and Applied Acarology, 80(3), 399-407.] but this is something that still needs to be addressed.
We have noted in the discussion (paragraph 1) that the virulence of V. jacobsoni (i.e. population sizes) may also be part of the tolerance mechanism in this bee population.